# Structural Stabilization of Human Transthyretin by *Centella asiatica* (L.) Urban Extract: Implications for TTR Amyloidosis

**DOI:** 10.3390/biom9040128

**Published:** 2019-03-29

**Authors:** Fredrick Nwude Eze, Ladda Leelawatwattana, Porntip Prapunpoj

**Affiliations:** Department of Biochemistry, Faculty of Science, Prince of Songkla University, Hat Yai, Songkhla 90112, Thailand; fredrickeze@rocketmail.com (F.N.E.); ladda.l@psu.ac.th (L.L.)

**Keywords:** transthyretin, TTR amyloidosis, protein misfolding, neuroprotection, *Centella asiatica*, antioxidants, triterpenoids, phenolics, HPLC-MS

## Abstract

Transthyretin is responsible for a series of highly progressive, degenerative, debilitating, and incurable protein misfolding disorders known as transthyretin (TTR) amyloidosis. Since dissociation of the homotetrameric protein to its monomers is crucial in its amyloidogenesis, stabilizing the native tetramer from dissociating using small-molecule ligands has proven a viable therapeutic strategy. The objective of this study was to determine the potential role of the medicinal herb *Centella asiatica* on human transthyretin (huTTR) amyloidogenesis. Thus, we investigated the stability of huTTR with or without a hydrophilic fraction of *C. asiatica* (CAB) against acid/urea-mediated denaturation. We also determined the influence of CAB on huTTR fibrillation using transmission electron microscopy. The potential binding interactions between CAB and huTTR was ascertained by nitroblue tetrazolium redox-cycling and 8-anilino-1-naphthalene sulfonic acid displacement assays. Additionally, the chemical profile of CAB was determined by liquid chromatography quadruple time-of-flight mass spectrometry (HPLC-QTOF-MS). Our results strongly suggest that CAB bound to and preserved the quaternary structure of huTTR in vitro. CAB also prevented transthyretin fibrillation, although aggregate formation was unmitigated. These effects could be attributable to the presence of phenolics and terpenoids in CAB. Our findings suggest that *C. asiatica* contains pharmaceutically relevant bioactive compounds which could be exploited for therapeutic development against TTR amyloidosis.

## 1. Introduction

Misfolding and/or aggregation of proteins is at the heart of the etiopathogenesis of a group of debilitating disorders referred to as amyloidosis [1]. Many of these diseases such as Parkinson’s disease, Huntington’s, Alzheimer’s, and transthyretin (TTR) amyloidosis are still incurable. As a result, a lot of investigative effort is focused on developing effective, safe, and reliable therapeutic agents.

In humans, TTR amyloidosis is a group of proteinopathies triggered by partial unfolding, misfolding, aggregation, and accumulation of TTR into a spectrum of cytotoxic aggregates [2]. Under normal physiological circumstances, TTR is a homotetrameric protein that transports thyroid hormones and retinol A via its interaction with a holo-retinol-binding protein [3]. In mature adults, a late-onset acquired TTR amyloidosis also known as senile systemic amyloidosis (SSA) is caused by the accumulation of wild-type in organs. SSA affects at least 25% of octogenarians, accounts for more than 10% of heart failure with preserved fraction [4], and is associated with the development of congestive heart failure [5,6]. Accumulation of variant TTR precipitates familial TTR amyloidosis, such as familial amyloidosis cardiomyopathy (TTR-FAC) [5], familial amyloidotic polyneuropathy (TTR-FAP), and central nervous system selective amyloidosis (CNSA) [7]. Onset for familial TTR amyloidosis could occur within the second decade of life depending on the TTR mutant involved. TTR amyloidosis can be very progressive and debilitating with mortality within the first 10 to 15 years of onset [8]. Orthotopic liver transplantation, with several limitations including highly invasive, unsuitability of patients, lack of donors, etc., is still the primary form of disease-modifying therapy for most TTR-FAP patients. Recently, the disease-modifying drug, Tafamidis, was approved for treating TTR-FAP [9]. However, in a just-concluded long-term study, it was observed that the slow progression of neuropathy in TTR-FAP patients could not be prevented by Tafamidis [10]. Thus, it is pertinent to search for safe, more effective, and less invasive therapeutic alternatives.

Although the current understanding of the molecular pathophysiology underlying TTR amyloidosis is still incomplete, it is widely regarded that tetramer dissociation into monomers is the initial and consequential step in its amyloidogenesis [11,12]. Thus, preventing tetramer dissociation by enhancing its kinetic stability has been proposed as an effective therapeutic strategy [8]. The kinetic stabilization strategy involves the use of small-molecule ligands to prevent conformational excursions of the native tetramer by binding to its thyroxine (T4)-binding sites [8]. Kinetic stabilization of TTR tetramers formed the rational basis for the development of the TTR-FAP drug, Tafamidis [13].

Native wild-type TTR, while significantly stable under normal physiological conditions [14], is susceptible to oxidative modifications, which could alter its conformation and stability [15]. The role of protein oxidation was noted in the increased amyloidogenicity of nitric oxide-modified TTR but not in unmodified TTR [15]. Also, age-associated oxidation of TTR reportedly impacts the onset of senile systemic amyloidosis [16]. Furthermore, while prefibrillar and mature TTR amyloid fibrils were not or less cytotoxic, all forms of oxidized native TTR were shown to be cytotoxic [2,16]. This backdrop highlights the role of oxidative damage in TTR amyloidogenesis and prompted the notion that, in addition to kinetic stabilizers, antioxidant treatment could be considered as an alternative/supplement therapy. Interestingly, most of the natural bioactives that have been reported to stabilize TTR such as quercetin, epigallocatechin gallate (EGCG), gallic acid, curcumin, and propolis extract are also potent antioxidants [17,18].

In addition, it has been noted that the protective effect exerted by antioxidants in chronic and neurodegenerative diseases in vivo is derived mainly from the additive, synergistic, and complementary effects of the several bioactive constituents present in the phytocomplex, such as plant extracts, fruits, and vegetables, rather than individual phytocomponents [19]. The antioxidant activity of medicinal plant extracts was also found to strongly correlate with their ability to limit the assembly of beta-amyloid, which is central to Alzheimer’s disease pathology [20]. For a multifactorial disease with a complex pathology such as TTR amyloidosis, plant extracts or products could provide potential multi-target therapeutic agents.

*Centella asiatica* is a small, succulent, and herbaceous plant popular in several parts of the world for its medicinal and culinary value. It is indigenous to the tropical and subtropical regions of Asia, Africa, and the southern parts of the USA. In Thailand, it is known locally as Bua-bok and is used for making a very popular herbal drink, Nam Bai Bua Bok. The fresh aerial parts are eaten with rice and are part of many local food recipes. In folk medicine, *C. asiatica* is a popular nervine and adaptogen. It is used for wound healing, treatment of neurological disorders, and promoting general wellbeing [21]. The major bioactive components of *C. asiatica* are a group of pentacyclic triterpenoids known as centellosides. In addition, *C. asiatica* is richly endowed with phenolics and flavonoids [22]. *C. asiatica* reportedly has many pharmacological and biological effects including antioxidant, anti-inflammatory, inhibition of β amyloid peptide aggregation and toxicity, and anti-α-synuclein aggregation [23,24]. Despite these reports, currently, there is no data on its potential pharmacological effect on TTR amyloidogenesis. Together with its rich phytochemical and good safety profile, we decided to investigate the potential role of *C. asiatica* in the modulation huTTR amyloidogenesis.

Thus, we examined the impact of a hydrophilic fraction of *C. asiatica* (CAB) on native huTTR structural stability and fibril formation using urea/acid-mediated denaturation assays, and transmission electron microscopy, respectively. We also determined the plausible binding interactions of CAB-huTTR complex and the chemical properties CAB. The present results provide relevant insight into the neuroprotective potential of *C. asiatica*, specifically, with regards to TTR amyloidosis.

## 2. Materials and Methods

### 2.1. Purification of huTTR from Plasma

Human TTR from plasma was isolated in a two-step protocol which consisted of a decrease in albumin burden and followed by preparative discontinuous native-PAGE as previously described [25,26]. The eluting fractions were monitored for the presence of huTTR by 10% native-PAGE. Protein bands were detected by silver staining. Fractions containing only huTTR were pooled, concentrated, and checked for purity by SDS-PAGE with Coomassie blue R-250 staining. The concentration of purified huTTR was determined by the Bradford assay [27] and stored at −20 °C.

### 2.2. Preparation of CAB

*C. asiatica* was obtained locally in Hat Yai city, Southern Thailand. The identity of the whole plant specimen was authenticated by Associate Professor Dr. Kitichate Sridith, Curator-in-Chief of the National Herbarium at Prince of Songkla University, Hat Yai, Thailand. A specimen was deposited in the herbarium with voucher number F.N.1 (PSU). Plant aerial parts were repeatedly washed with tap water followed by reverse osmosis water. The plant sample was air-dried for 12 h to reduce moisture content and then oven-dried at 60 °C for another 12 h. Dried *C. asiatica* was ground into a fine powder and stored in an opaque container at −20 °C for extraction within 24 h.

*C. asiatica* powder (450 g) was extracted with 2 L of cold acetone/methanol/water (2:2:1 *v*/*v*/*v*) containing 0.5% glacial acetic acid for 2 h as previously reported [28]. The plant mixture was filtered through a Buchner funnel overlaid with filter paper (Whatman No. 1) using a vacuum pump. The filtrate was concentrated to one-third of its original volume by a rotary evaporator at a temperature of 40 °C. Thereafter, the filtrate was defatted by partitioning in an equal volume of n-hexane twice, using a separatory funnel. The lower phase was further partitioned in dichloromethane twice to remove chlorophyll, waxes, and other less polar components. The upper hydrophilic phase in the separatory funnel was collected and residual organic solvents were removed under vacuum by speed-vac to remove residual organic solvents, and then lyophilized to a hygroscopic powder. This powder obtained from the hydrophilic fraction of *C. asiatica* was hereafter referred to as CAB (*C*. *asiatica* bioactives). CAB was aliquoted into opaque vials and stored at −20 °C. The scheme for CAB preparation is shown in Appendix A.

### 2.3. Nitroblue Tetrazolium (NBT) Redox-Cycling Assay

The binding of CAB to huTTR was determined by NBT staining which distinguishes quinone-modified from unmodified proteins [29]. Human TTR (2.1 µg/µL) in 50 mM Tris-HCl pH 7.5 was incubated in the presence of CAB or gallic acid (GA), or DMSO (vehicle), at 10× the molar equivalent of human TTR concentration. The samples were mixed with sample buffer containing 4% SDS and immediately boiled for 10 min prior to separation by SDS-PAGE (15% resolving gel). The gel was electrotransferred onto a nitrocellulose membrane. After transfer, the membrane was stained with Ponceau S dye (0.1% Ponceau S in 5% acetic acid) for 1 h to confirm blotting. Subsequently, the membrane was washed with distilled water, Tris-buffered saline with Tween 20 (TBS-T) and rinsed with Milli-Q water to remove the Ponceau S stain. Then, it was re-stained with glycinate/NBT solution (10 mg NBT tablet in 14 mL of 2 M potassium glycinate buffer, pH 10) for 45 min to identify proteins that interacted with phenolics or related compounds.

### 2.4. Determination of the Stability of huTTR in the Presence of CAB

TTR tetramer dissociation into monomeric subunits is quite slow under normal physiological pH. However, dissociation of TTR tetramer and subsequent misfolding of monomers is significantly increased in vitro under conditions of high urea concentration or mild acidity [14]. Resistance to urea-induced or acid-induced dissociation of TTR tetramer to monomers provides insight into the stability of native TTR against the partial denaturation of monomers required for amyloidogenesis. This is because tetramer dissociation precedes monomer unfolding and misfolding [30]. Purified huTTR in 50 mM Tris-HCl buffer pH 7.5 was dialyzed against 10 mM sodium phosphate buffer containing 100 mM KCl and 1 mM ethylenediamine tetraacetic acid (EDTA), pH 7.4 (GF buffer) for 24 h with 3 buffer changes prior to stability assays as follows.

#### 2.4.1. Against Urea-Mediated Denaturation

The influence of CAB on huTTR stability against urea-mediated dissociation was evaluated as previously described [31]. In brief, huTTR (1 µg) in a GF buffer, pH 7.4 was supplemented with CAB (100× molar equivalents with respect to huTTR) dissolved in GF buffer containing DMSO, 50:50 *v*/*v* (GMSO) or GMSO only, and incubated for 2 h at 37 °C. Denaturation was triggered by adding urea (in GF buffer containing 1 mM 1,4-Dithiothreitol, pH 7.4) into the protein complex to a final concentration of 7 M and subsequently incubating for 96 h at ~6 °C in the dark. Folded huTTR (i.e., tetramers, trimers, and dimers) that remained after the denaturation was determined by 14% Tricine SDS-PAGE. The running buffer contained 0.025% SDS and the sample loading buffer 0.2% SDS. It has been suggested that these detergent concentrations are enough to prevent monomer reassociation, but too low to trigger tetramer dissociation [32]. Resolved proteins were identified by staining with Coomassie brilliant blue R-250 dye (0.1%) solution. Bands representing folded huTTR on the gel were quantified by densitometry using gel documentation on LabWorks 4.0 software (UVP Ltd., Cambridge, UK). The relative percentage of folded huTTR that remained after denaturation was calculated as an indication of native huTTR stability.

#### 2.4.2. Against Acid-Mediated Denaturation

The effect of CAB on the stability of huTTR under mildly acidic conditions was performed as described previously [33]. Briefly, huTTR (0.5 µg/uL) in a GF buffer, pH 7.4 was supplemented with or without 100× molar excess of CAB followed by incubation for 2 h at 37 °C. Tetramer dissociation was initiated by reducing the pH of the reaction mixture to 4.0 by adding an equal volume of acetate/acetic acid buffer containing 100 mM KCl, 1 mM EDTA, and 2 mM dithiothreitol (DTT), pH 4.0. Thereafter, the protein complex was incubated at 37 °C for 14 days in the dark and under aseptic condition. Glutaraldehyde cross-linking assay was performed to determine the amount and quaternary structure of huTTR that remained after 14 days of denaturation stress. Briefly, glutaraldehyde was added to the protein complex to a final concentration of 2.5%, and the assay mixture was incubated for 4 min at room temperature. Cross-linking was terminated by the addition of NaBH_4_ (7% in 0.1 M NaOH) at an equal volume to glutaraldehyde. Then, the mixture was mixed with 4x loading buffer containing 8% SDS and 5% beta-mercaptoethanol and boiled for 10 min prior to analysis by 10% Tricine SDS-PAGE. The separated protein bands were visualized by Coomassie blue staining. The intensity of the tetrameric form of huTTR was quantified by densitometry using gel documentation. The stability of huTTR was inferred from the relative percentage of tetramer that remained after denaturation.

### 2.5. Determination of 8,1-ANS Binding Displacement from huTTR by CAB

Fluorophore, 8-anilino-1-naphthalene sulfonic acid (8,1-ANS), reportedly binds at the T4-binding site of native huTTR in solution [34]. Thus, its displacement by small-molecule ligands is often used to probe for ligand binding to the T4-binding site of TTR tetramer [33]. To perform 8,1-ANS binding-displacement assay, huTTR (0.055 µg/µL) was incubated in the presence or absence of 8,1-ANS, (10 µM) for 10 min, followed by the addition of CAB at 25× and 50× molar equivalents with respect to human TTR. To measure fluorescence, protein samples were excited at 385/40 and emission was collected at 480/20 using a Synergy HT (Bio-Tek Instruments, Winooski, VT, USA) microplate reader.

### 2.6. Determination of the Influence of CAB on huTTR Fibrillation by Transmission Electron Microscopy (TEM)

TEM was used to study the impact of CAB on huTTR fibril formation as previously described [35]. Human TTR (1 µg/µL) in a GF buffer was supplemented with or without CAB (20× molar equivalents with respect to huTTR). The mixture was subsequently incubated for 2 h at 37 °C to facilitate binding. Fibril formation was initiated by reducing pH to 4.0, via the addition of an equal volume of acetate buffer (pH, 4.0) to the mixture, and followed by incubation at 37 °C for 7 days. The extent of aggregate and fibril formation was ascertained by TEM of the TTR mixtures. In brief, aliquots of the TTR mixtures were diluted with Milli-Q water (1:20, *v*/*v*). A drop (~10 µL) of the solution was applied onto a 400-mesh formvar-coated copper grid for 1 min. Excess sample was removed with a wedge of filter paper. The grid was immediately rinsed with a drop of Milli-Q water, dried, and negatively stained with 2% uranyl acetate in 70% methanol for 2 min. Excess stain was blotted out and the grid was thoroughly air-dried. TEM images were obtained with a JEM-2010 (JEOL Ltd., Akishima, Tokyo, Japan) electron microscope, running at 160 KV.

### 2.7. Determination of CAB Antioxidant Properties

#### 2.7.1. DPPH (1,1-Diphenyl-2-picrylhydrazyl) Radical Scavenging Activity

CAB was tested for its scavenging effect on DPPH radical following the method as described [36]. CAB was prepared in 50% aqueous methanol while Trolox (purchased from Aldrich Chemical Company, Steinheim, Germany) was prepared in distilled water and served as an antioxidant standard. CAB or Trolox solution (10 µL) was diluted with methanol (140 µL), and then DPPH solution in methanol (150 µL, 0.1 mM) was added. The reaction mixture in which CAB was replaced with aqueous methanol and Trolox was replaced with distilled water were included as controls. Blank probes contained methanol (290 µL) and 10 µL of CAB or Trolox. Blank probes for controls contained only methanol (300 µL). Sample and control reactions were incubated for 30 min before reading absorbance at 515 nm. After blank correction, DPPH radical scavenging activity (%) was obtained from the equation:inhibition (%) = (absorbance of control − absorbance of sample)/absorbance of sample × 100(1)

A curve was plotted for inhibition versus concentration. The antioxidant activity of CAB and Trolox was obtained from the curve and expressed as the concentration that scavenged the DPPH radical by 50% (IC_50_) in a mean ± standard deviation of triplicates determination.

#### 2.7.2. Ferric Reducing Antioxidant Power (FRAP)

The reducing power of CAB was determined by FRAP assay as described by [37]. FRAP solution containing 300 mM acetate buffer pH 3.6, 10 mM TPTZ 2,4,6-Tri(2-pyridyl)-s-triazine (TPTZ) solution, and 20 mM FeCl_3_.6H_2_O solution (10:1:1, *v*/*v*/*v*) was prepared and incubated for 30 min at 37 °C. CAB solution was prepared by diluting the powder with 50% aqueous methanol. Trolox (up to 600 µmol/L) was included as the antioxidant standard for calibration. The assay was performed in a 96-well plate. Aliquots (10 µL) of CAB solution or Trolox solution were mixed with FRAP solution (200 µL) and incubated for 30 min at 37 °C. Then, absorbance was monitored at 650 nm. CAB antioxidant activity was expressed as a micromole of Trolox equivalents per gram of CAB. The higher the FRAP value, the stronger the reducing potential.

### 2.8. Determination of the Chemical Composition of CAB

#### 2.8.1. Total Phenolic Content

The phenolic content in CAB was determined using the Folin–Ciocalteu assay with slight modification [38]. CAB solution (10 µg/µL) and gallic acid (GA; 1 µg/µL) in DMSO/methanol (10:90, *v*/*v*) were prepared in which the latter was used as a standard. Aliquot (100 µL) of CAB solution, blank or standard solution (from 0 to 80 µg), was added into test tubes. Then, 200 µL of Folin–Ciocalteu reagent (10%) was added and vortexed. After 5 min of incubation, 800 µL of Na_2_CO_3_ (700 mM) was added and vortexed. The assay mixture was incubated in the dark at room temperature for 2 h. Then, 200 µL of the blue mixture formed was transferred into a 96-well microplate and absorbance was measured at 765 nm, using a Synergy HT microplate reader (Bio-Tek Instruments, Winooski, VT, US). A plot of the amount of gallic acid (µg) versus absorbance at 765 nm was prepared and used to obtain the total phenolic content of the CAB. Linearity for the gallic acid standard curve for determining phenolic content was obtained between 0 and 20 µg. Data obtained from quadruplicates were reported as mean ± standard error of the mean and expressed in milligrams of gallic acid equivalent per gram of CAB.

#### 2.8.2. Total Flavonoid Content

Total flavonoid content of CAB was determined by the aluminum chloride colorimetric assay with slight modifications as previously described [39]. CAB solution (10 µg/µL) was prepared in 50% aqueous methanol, and quercetin standard (1 µg/µL) was prepared in 100% methanol. CAB or quercetin solution (30 µL) was diluted with 160 µL of methanol and subsequently mixed with 10% aluminum chloride (30 µL) and 1 M sodium acetate (30 µL) in a total volume of 1100 µL of the assay. The mixture was vortexed and subsequently incubated in the dark for 30 min. Absorbance was read at 415 nm. Sample and standard solutions were prepared in quadruplicates. The calibration curve of the absorbance versus concentration of quercetin standard was plotted and used to calculate total flavonoid content of which was expressed as the mean ± standard error in milligrams of quercetin equivalents per gram of CAB.

#### 2.8.3. Thin-Layer Chromatography (TLC) Profile of CAB

To determine the major phytoconstituents present in CAB, TLC was conducted on aluminum-backed silica gel 60 NH_2_ F_254S_ TLC plates (Merck Millipore, Darmstadt Germany). CAB was reconstituted in methanol/water/acetic acid (100:10:4, *v*/*v*/*v*) and 5 µL (~300 µg) of CAB solution was spotted on a TLC plate, and the plate was transferred to a glass jar lined with filter paper and saturated for 30 min with developing solvent containing water/acetic acid/formic acid/ethyl acetate (5:1:1:17, *v*/*v*/*v*/*v*). The separation was terminated after the solvent front had migrated about 8 cm from the origin, and the plate was dried up in a fume hood to remove any residual solvent. One plate was stained with *p*-Anisaldehyde-sulfuric acid reagent, another was dipped into 3% ferric chloride reagent, and the third was sprayed with 2% AlCl_3_ in methanol after saturation with ammonia solution. All plates were observed under visible light and ultraviolet light (254 and 366 nm) before and after derivatization, respectively.

#### 2.8.4. High-Performance Liquid Chromatography-Mass Spectrometry (HPLC-MS) Fingerprint of CAB

LC-MS analysis of CAB was performed as previously described [40]. The sample was resuspended in methanol and filtered through a 0.22 µm nylon membrane. Separation of analyte was performed with a Poroshell 120 EC-C18 column (4.6 × 150 mm, 2.7 µm), Agilent Technologies. The solvent system consisted of water (as eluent A) and acetonitrile (as eluent B). Both eluents contained formic acid (0.1% *v*/*v*). The flow rate was maintained at 0.3 mL/min. The gradient employed for the separation was as follows: 0% B, 0–5 min; then increased linearly to 80% B, 5–50 min; maintained at 80% B, 50–53 min; then returned to 0% B, 53–55 min; and finally, re-equilibrated in 0% B, 55–60 min. The stream of separated components was channeled into a micrOTOF-QII™ESI-Qq-TOF mass spectrometer (Bruker Daltonics, Bremen, Germany) equipped with an electrospray ionization source. Ions were detected in the negative mode within a mass range of 50–1500 Da. Parameters set for mass spectra acquisition included a capillary voltage of 3500 V, nebulizer pressure of 2.0 Bar, drying gas of 8.0 L/min, and drying temperature of 180 °C. Sodium formate solution (10 mM) was used as a calibration standard to ensure that the mass/charge ratios recorded by the instrument were indeed accurate. Data acquisition was performed on Bruker Compass DataAnalysis 4.0 software (Bruker Daltonics, Bremen, Germany). Tentative identity of compounds was established by comparing the accurate mass measurements obtained with previously identified compounds in *C. asiatica* reported in literature and online databases including METLIN and ChemSpider.

### 2.9. Statistical Analysis

Data of the huTTR stability assays were presented as the mean ± standard error of the mean (SEM). The percentages of folded huTTR or tetramers preincubated with or without CAB that remained after denaturation stress were compared using ANOVA followed by the Tukey–Kramer test. Differences were considered statistically significant at *p* < 0.05.

## 3. Results

### 3.1. CAB Directly Binds to huTTR

To determine whether CAB binds directly to huTTR purified from human plasma (Appendix A), we used nitroblue tetrazolium (NBT) redox-cycling staining [35]. Some natural product components, especially phenolics and related compounds, reportedly auto-oxidize into quinones which can interact with nucleophilic centers in protein such as the sulfhydryl group of a cysteine residue [41]. Under alkaline pH conditions (for example, in the presence of NBT/potassium-glycinate solution, pH 10), such quinoprotein adducts formed undergo redox cycling with glycine accompanied by a concomitant reduction of NBT into a visible blue-purple formazan color on the membrane [29]. This color formation indicates interaction between the bioactives and protein.

As shown in Figure 1a, Ponceau S staining revealed the presence of huTTR monomers and dimers based on their positions on the membrane. TTR tetramer normally migrates as an apparent dimer under non-reduced and non-boiled SDS-PAGE. [42,43]. The membrane was later distained of Ponceau S. Upon subjecting the membrane to NBT/glycinate staining, purple huTTR bands were detected in the lanes containing CAB, whereas no color reaction was observed in the absence of CAB (DMSO only) (Figure 1a). In the presence of gallic acid (positive control), the NBT stain produced a bright formazan color corresponding to the band position of the huTTR monomer. However, in the presence of CAB, the formazan color was present at the position representing huTTR dimers and monomers (although, quite faintly) (Figure 1a). The presence of the formazan color in the lane of CAB-associated huTTR indicated direct binding interactions. The higher intensity of the formazan color associated with the dimeric band suggested CAB binding preference for quaternary over tertiary huTTR forms, and that the CAB–huTTR complex formed is quite stable under reduced SDS-PAGE conditions.

### 3.2. CAB Binds to the T4-Binding Sites of huTTR

To further understand the binding interactions between huTTR and CAB, 1,8-ANS binding displacement assay was performed. The fluorescent dye, 8,1-ANS is very sensitive to its surrounding chemical environment [44]. In aqueous environments, it produces a weak fluorescence; however, the fluorescent quantum yield increases significantly with a blue shift in the emission maxima when the surrounding becomes less polar [44]. ANS binds to the T4-binding sites of TTR tetramers with a corresponding increase in the magnitude of fluorescent quantum yield and a blue shift from 515 nm to about 465 nm [34]. Given the hydrophobic nature of the T4-binding pockets, the increase in fluorescence due to 8,1-ANS binding makes sense. Thus, several studies have utilized the displacement of 8,1-ANS from the T4-binding site as a probe to determine the binding of small-molecule ligands to the T4-binding pockets of tetrameric TTR [18,33]. In addition, the ANS fluorescence could serve as an indicator for the stability and compactness of the TTR core [45]. In this study, the binding of ANS to huTTR increased the quantum yield (Figure 1b). The addition of CAB reduced the fluorescent intensity dose-dependently, which suggests the displacement of 8,1-ANS from the T4-binding site of huTTR tetramers (Figure 1b). These results suggest the binding of CAB at the T4-binding sites of huTTR tetramers.

### 3.3. CAB Increases huTTR Structural Stability

To determine the impact of CAB on huTTR kinetic stability in vitro, urea-mediated denaturation assay was performed. HuTTR was preincubated with or without CAB, and subjected to denaturation by 7 M urea (final concentration). The intensity of folded huTTR was higher in the presence of CAB than in its absence (Figure 2a). The percentage of folded CAB-associated huTTR that resisted urea-mediated dissociation was 89.65 ± 4.6%. In contrast, the percentage of folded huTTR in the absence of CAB was lower, 58.87 ± 4.37% (Figure 2b). These results demonstrated that CAB improved the stability of huTTR tetramer against the denaturation stress.

To further understand the potential role of CAB on huTTR amyloidogenesis, we examined the quaternary structural changes in CAB-associated huTTR under urea denaturation by glutaraldehyde cross-linking assay. Human TTR was preincubated without or with varying concentrations of CAB and subjected to denaturation stress. Tricine SDS-PAGE of glutaraldehyde cross-linked huTTR samples revealed protein bands of ~60 kDa and ~16 kDa, corresponding to huTTR tetramers and monomers, respectively (Figure 2c). It is noteworthy that while the intensity of the monomeric band diminished as the amount of CAB increased, the intensity of huTTR tetrameric band increased correspondingly (Figure 2c). These findings indicated that CAB increased the quaternary structural stability of huTTR concentration-dependently.

In vitro, TTR rapidly dissociates when subjected to mildly acidic conditions. This facilitates its auto-aggregation [11,46]. In order to understand the impact of CAB on the stability of huTTR under conditions which promote amyloidogenesis, acid-mediated denaturation assay was performed. After subjecting huTTR samples to denaturation stress, the percentage of huTTR tetramers in the presence of CAB was higher than in the absence (Figure 3). In the presence of CAB, the percentage of huTTR tetramers that remained was 68.32 ± 0.81% as opposed to 30.24 ± 0.78% in its absence. These results indicated that CAB enhanced the quaternary structural stability of huTTR and confirmed the findings from the urea-mediated denaturation assay.

### 3.4. CAB Prevents huTTR Fibril Formation

The influence of CAB on huTTR fibrillation was determined by examining the morphology of the protein using transmission electron microscopy (TEM). Based on the electron micrographs, no huTTR mature fibril was formed in the presence of CAB; however, the formation of aggregate was not impeded (Figure 4b). Conversely, in the absence of CAB, not only did huTTR transform into aggregate, but also produced mature fibrils (Figure 4c,d). These findings suggested that CAB prevented the formation of mature huTTR fibrils under moderately acidic conditions in vitro. The electron micrograph of CAB incubated alone under similar conditions is shown in Appendix A.

### 3.5. Antioxidant Activity of CAB

To determine the antioxidant activity of CAB, DPPH radical scavenging and FRAP assays were performed. The DPPH assay measures the ability of component(s) to quench the DPPH radical by transferring H to it. From the DPPH assay, CAB and the antioxidant standard displayed IC_50_ values of 28.53 and 2.81 µg/mL, respectively. While the radical-scavenging effect of CAB was lower than that of the standard, the abilities of both substances to remove the free radical were of the same magnitude. The FRAP assay showed that the reducing potential of CAB was 284.17 µmol Trolox equivalents per gram of CAB which is considered high [47]. These results demonstrated that CAB possessed potent antioxidant capacity.

### 3.6. Chemical Characterization of CAB

#### 3.6.1. Total Phenolic and Flavonoid Contents

The antioxidant activities of many fruits and plants have been largely attributed to the presence of phenolics and flavonoids [48]. In this report, Folin–Ciocalteu and aluminum chloride colorimetric assays were used to determine the total phenolic and flavonoid contents of CAB, respectively. The content of phenolics in CAB was 43.86 ± 0.34 mg of gallic acid equivalent per gram, while the flavonoids content was 13.89 ± 0.23 mg of quercetin equivalents per gram of CAB.

#### 3.6.2. TLC Profile of CAB

Phytochemical analysis to determine the major chemical profile of CAB was performed by thin layer chromatography on an aminopropyl-modified silica gel TLC plate. After derivatization with *p*-Anisaldehyde-sulfuric acid reagent, emergence of pink, purple, and yellow bands were observed under white light, indicating the presence of terpenoids and phenolics (Appendix A). The emergence of bands that quenched the fluorescence indicator on the TLC plate under ultraviolet light at 254 nm (Appendix A), suggested the presence of phenolics. Likewise, the appearance of bright yellow colored bands under white light, after the TLC plate was saturated with ammonia solution and sprayed with 2% methanolic AlCl_3_, (Appendix A) was typical of phenolic acids and flavonoids [49].

#### 3.6.3. HPLC-QTOF-MS Analysis of CAB

HPLC-QTOF-MS was used to determine the chemical profile of CAB. The base peak chromatogram (negative ion mode) is revealed in Figure 5. The tentative identity of the major peaks obtained, retention times, accurate masses, and predicted molecular formulae are presented in Table 1. Identification of compounds was accomplished by comparing the accurate masses obtained with those of previously reported compounds present in *C. asiatica* found in the literature [40,50] and in online databases including METLIN and ChemSpider. Eight caffeoylquinic acids (chlorogenic acids) were identified including 3-*O*-caffeoylquinic acid, 5-*O*-caffeoylquinic acid, 3,4-*O*-dicaffeoylquinic acid, 3,5-*O*-dicaffeoylquinic acid, 3,5-*O*-dicaffeoyl-4-malonylquinic, 3-caffeoyl-4-feruloylquinic acid, and 4,5-*O*-dicaffeoylquinic acid (Table 1). In addition, seven pentacyclic triterpenoids were present including asiaticoside B, madecassoside, centellasaponin B, centellasaponin A, asiaticoside, avenacoside A, and soyasaponin I. The flavonoids quercetin 3-*O*-glucuronide and eriodictyol 7-(6-galloylglucoside) were also identified (Table 1). Thus, the prevalent chemical groups in CAB were chlorogenic acids and pentacyclic triterpenoids. The chemical composition of CAB was similar to those previously reported for *C. asiatica* [22,40,50].

## 4. Discussion

In this study, we demonstrated that the bioactive compounds in CAB modulated huTTR amyloidogenesis by binding to the T4-binding sites of the tetramer. This enhanced the quaternary structural stability of huTTR against denaturation stresses. In addition, CAB prevented the fibrillation of huTTR under acid-induced aggregation conditions and demonstrated potent antioxidant activity.

Biophysical evidence has shown that dissociation of TTR tetramer is the initial and most critical step in TTR amyloidogenesis [8,11,12,51,52,53]. Thus, it is reasoned that enhancing TTR tetramer kinetic stability would make for an effective therapeutic strategy. Kinetic stabilization of native TTR tetramers can be accomplished by small-molecule ligands binding to the T4-binding sites of tetrameric TTR, which elevates the kinetic barrier for its dissociation [8]. Since TTR amyloidogenesis is a concentration-driven process, by enhancing tetramer stability, the process is depleted of required amyloidogenic-competent monomers [54].

Natural products such as curcumin and propolis are now receiving more attention as potential kinetic stabilizers [18,55]. Commercial bee propolis (brand name: Bio30) extract, rich in caffeoyl acid phenethyl ester, reportedly stabilized the tetramers and inhibited transthyretin amyloidogenesis in vitro [18]. Our data demonstrated that CAB preserved the quaternary structural stability of huTTR, as reflected by the resistance of CAB-associated huTTR against urea- and acid-mediated denaturation conditions (Figure 2 and Figure 3). These results concurred with previous findings of increased TTR structural stability by natural products [18,35,55,56]. This often involves, as earlier noted, binding of the small-molecule ligands at the T4-binding sites of the homotetramers [17]. The two T4-binding sites are situated in a particularly weak dimer–dimer interface. Electrostatic repulsion by side chains of symmetric residues (e.g., Lys 15, Lys15′) located at the surface of this hydrophobic pocket contributes significantly to the decrease in tetramer stability [53]. Since more than 99.5% of the T4-binding sites of TTR tetramers in serum are unoccupied under normal physiological conditions [57], an opportunity is created for small-ligand binding. Binding of small-molecule ligands at the T4-binding sites within the hydrophobic channel increase interactions and serve as tithers between the two dimers, and as potential anion shields that reduce the effects of electrostatic repulsion between the dimers. Consequently, the conformational stability of the tetramer is enhanced [53,56]. Thus, the increase in huTTR stability in this study could be attributed to the binding of CAB as small ligands at T4-binding sites of the tetramers as revealed by the ANS-binding displacement assay (Figure 1b). The increased ANS fluorescence (Figure 1b) is also indicative of the compactness of tetramers and integrity of the protein inner core [45] upon CAB binding. This was further supported by the preference of CAB binding to folded huTTR as revealed by the NBT assay (Figure 1a). CAB is phenol-rich. It is plausible that the phenolic components present in CAB form quinoprotein adducts with huTTR via Schiff base addition involving the Lys15, Lys15′ residues within the T4-binding sites [58,59]. Details of the molecular interactions between CAB and huTTR could constitute a future investigation.

CAB also binds (albeit, weakly) to huTTR monomers as indicated by the faint formazan color in the NBT redox-cycling assay (Figure 1a). Non-native monomers are the precursors for TTR aggregation and fibril formation [30,51]. The binding of CAB to huTTR monomers could partly explain the inhibition of huTTR amyloid fibril formation under conditions of mild acidity (Figure 4). Previously, it was reported that the aqueous extract of *C. asiatica* completely prevented the formation of amyloid-beta aggregates from monomers, and disintegrated preformed fibrils [23]. While our data showed that CAB prevented the formation of mature huTTR amyloid fibrils, the formation of aggregates was unabated (Figure 4b). This could be because huTTR oligomers, after being modified by CAB could no longer proceed in the amyloid cascade, (i.e., form mature fibrils). While such aggregates could be innocuous as previously demonstrated [60] in some natural phenolic products, it is yet to be investigated in CAB. The LC-MS data (Table 1) revealed that the major group of compounds in CAB are triterpenoids and phenolics. The ability of CAB to modulate TTR amyloidogenesis and fibril formation could be partly explained by the aromatic rings present in these compounds, and the formation of noncovalent interactions between them and amino acid residues of the β-sheet-rich core of the amyloidogenic protein [18,23,61].

Pro-oxidation is a feature in the pathophysiology of many neurodegenerative diseases including TTR amyloidogenesis [15,16]. Cytotoxic TTR comes in various conformations including amyloidogenic monomers, small soluble aggregates, and oxidatively-modified tetramers [16]. Upon interaction with cells, amyloidogenic TTR reduces cell viability and induces the release of reactive species [62]. Oxidatively modified TTR reportedly influences amyloidogenicity and age of onset in senile systemic amyloidosis [16]. In addition, a decrease in serum albumin antioxidant capacity was reported to accelerate TTR amyloid deposition in FAP patients [63]. Contrariwise, administration of a potent antioxidant, carvedilol, decreased TTR deposition, and oxidative and ER stresses in a FAP animal model [64]. All this underscores the importance of pro-oxidants in the promotion of antioxidants in amelioration of TTR amyloidosis. Our data indicated that CAB possessed potent radical scavenging activity and reduction ability as demonstrated by IC_50_ and FRAP values 28.53 µg/mL and 284.17 µmol Trolox equivalents per gram of CAB, respectively. Compared with the previous reports on 44 plant extracts [47,65], CAB possessed excellent antioxidant capacity which could be attributed to its high phenolic and flavonoid content [48,66] (Pittella et al., 2009). Thus, the mitigation of huTTR amyloidogenesis by CAB could partly be attributed to its phenolic content [35,60,61] and antioxidative ability [20,64,67]. In addition, CAB as an antioxidant could potentially enhance oxidative balance by quenching reactive species and thus, reducing their deleterious effects on not only, huTTR, but also lipids, genetic materials, and aggregated-TTR-induced oxidative stress [48,64].

A potential drawback of this study is the possibility that a single compound or group of compounds might be responsible for the observed bioactivity of CAB. However, it has been widely reported that the bioactivity exerted by phytochemicals in natural products such as plant extract is mainly due to the additive, synergistic, and complementary effects of many rather than any individual phytoconstituent [19] (Liu, 2003). Also, CAB as a phytocomplex, rather than an isolated phytochemical, could make for a better multitarget agent for a complex multifactorial disorder such as TTR amyloidosis.

In this report, we have provided strong evidence that the hydrophilic fraction of *C. asiatica* contains potentially useful bioactives or lead compounds that could ameliorate TTR amyloidosis. Since TTR stability is vital in beta-amyloid clearance [68], CAB could also be relevant in Alzheimer’s disease therapy. Prior, evidence had been presented for the neuroprotective and cognitive enhancement action of *C. asiatica* extracts [69,70]. In addition, *C. asiatica* bioactives were shown to be neuritogenic [71], synaptogenic [70], and anxiolytic [72]. Moreover, previous studies indicated that *C. asiatica* extracts decreased oxidative damage [73], increased cyclic AMP response element binding protein (CREB) phosphorylation, modulated the extracellular signal-regulated kinases (ERK) and protein kinase B signaling pathways [69,71] and was anti-inflammatory [74], which suggests a wide spectrum of biological activities. However, pertaining to the anti-TTR amyloidogenic activity of CAB, further studies are still required in order to elucidate the potency and safety in cell and animal models of the disease.

## 5. Conclusions

Based on the mechanistic studies presented herein, we determined that CAB might be a potential candidate or source material for future therapeutic development for TTR amyloidosis and/or related neurodegenerative diseases.

## Figures and Tables

**Figure 1 biomolecules-09-00128-f001:**
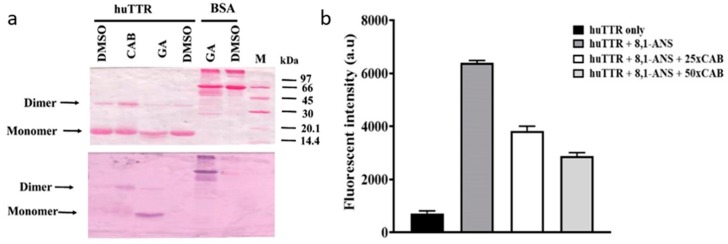
(**a**) Ponceau S staining of human transthyretin (huTTR) treated or non-treated (vehicle) with bioactives after electroblotting of SDS-PAGE onto the nitrocellulose membrane (above). Protein: Drug ratio of 1:10. Nitroblue tetrazolium (NBT)/glycinate staining of the membrane above after de-staining Ponceau S (below). (**b**) Displacement of bound 8-anilino-1-naphthalene sulfonic acid from huTTR by increasing concentrations of *Centella asiatica* (CAB). The bar chart columns and error bars represent the means of triplicates and standard deviations, respectively.

**Figure 2 biomolecules-09-00128-f002:**
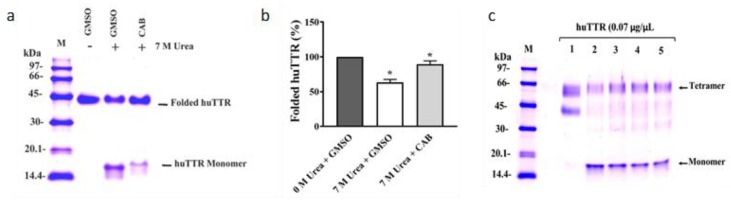
(**a**) Resistance to urea-mediated denaturation of huTTR in the presence and absence of *Centella asiatica* bioactive (CAB) components. HuTTR (0.07 µg/µL) was incubated in the presence of CAB (6.67 µg/µL) dissolve in GMSO or in the presence of GMSO only. (**b**) Bar chart represents the means of huTTR with and without CAB (protein: CAB mass ratio 1:100) and with their standard errors. * values are significantly different at *p* < 0.05. (**c**) Tricine SDS-PAGE gel (10%) of huTTR with increasing concentrations of CAB subjected to urea-denaturation stress and cross-linked with glutaraldehyde. M: Protein molecular weight marker; 1: 0 M urea + GMSO only; 2: 7 M urea + GMSO only; 3: 7 M urea + CAB (3.33 µg/µL); 4: 7 M urea + CAB (6.66 µg/µL); 5: 7 M urea + CAB (13.33 µg/µL).

**Figure 3 biomolecules-09-00128-f003:**
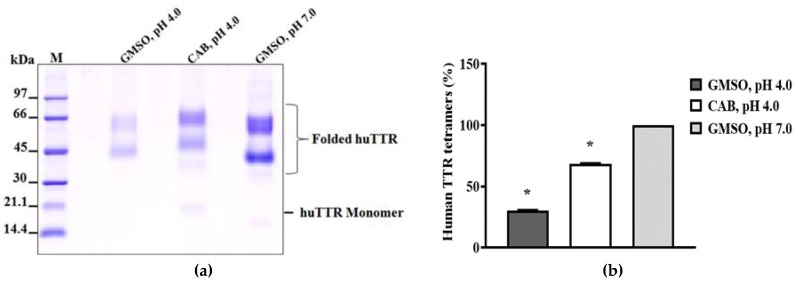
The resistance of human transthyretin in the absence or presence of CAB against acid-mediated denaturation. HuTTR (0.5 µg/µL) was incubated with or without CAB (50 µg/µL) and subjected to acidic denaturation conditions for 2 weeks. (**a**) The image represents the resolved protein mixtures on 10% Tricine SDS-PAGE gels after cross-linking with glutaraldehyde. M: Protein molecular weight marker; 1: huTTR with GMSO only, pH 4.0; 2: huTTR with CAB, pH 4.0; 3: huTTR with GMSO only, pH 7.0. (**b**) Bar chart represents the extent of huTTR stability, derived from the percentage of tetramers (%) left after denaturation. * values are significantly different at *p* < 0.05.

**Figure 4 biomolecules-09-00128-f004:**
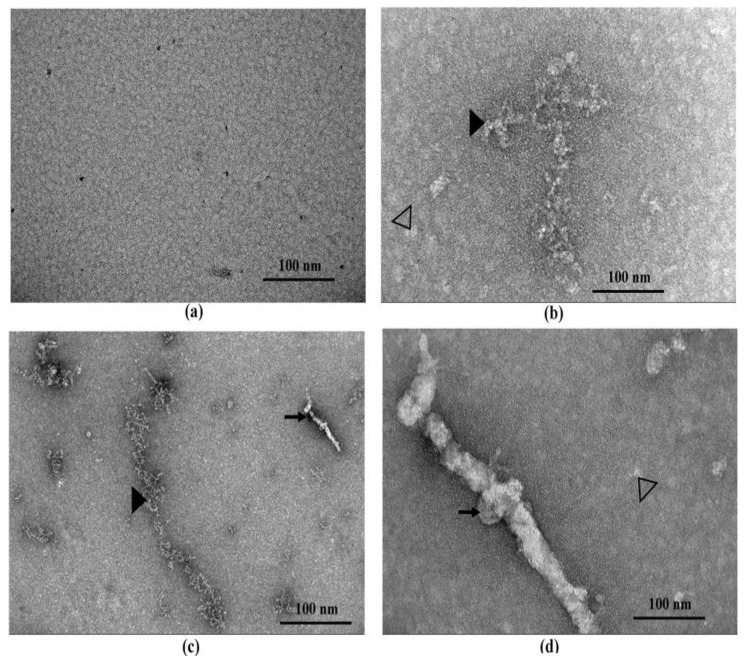
The TEM micrograph of huTTR incubated in the presence or absence of CAB. (**a**) Human transthyretin (TTR) in GF buffer, pH 7.4 supplemented with GMSO and incubated at −20 °C. (**b**) Human TTR supplemented CAB, final pH 4.0 and incubated at 37 °C for 7 days. (**c**) Human TTR supplemented with GMSO, pH 4.0 and incubated at 37 °C for 7 days. (**d**) Same image as (**c**) but with higher magnification. Arrows represent mature fibrils, filled arrow-heads large amorphous aggregates, and not-filled arrow-heads oligomers.

**Figure 5 biomolecules-09-00128-f005:**
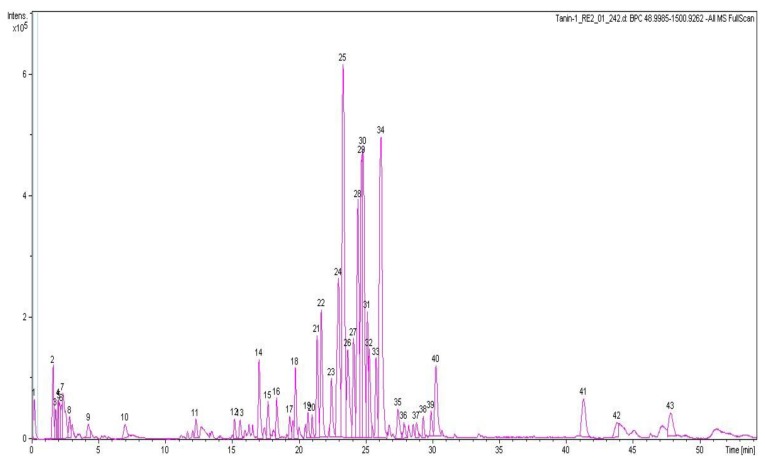
Base peak chromatogram of compounds in CAB obtained using HPLC-QTOF-MS in the negative mode.

**Table 1 biomolecules-09-00128-t001:** Profile of major bioactives detected in CAB by HPLC-QTOF-MS.

Peak No.	Retention Time	Observed Accurate Mass (*m*/*z*)	Predicted Formula	Calculated Mass (Da)	Tentative Identity of Compound
**4**	2.0	503.1527 [M − H]^−^	C_21_H_28_O_14_	504.1479	Caffeic acid dihexoside
**12**	15.1	353.0877 [M − H]^−^	C_16_H_18_O_9_	354.0951	3-*O*-Caffeoylquinic acid
**14**	17.0	353.088 [M − H]^−^	C_16_H_18_O_9_	354.0951	5-*O*-Caffeoylquinic acid
**21**	21.3	693.2790 [M − H_2_O − H]^−^	C_34_H_48_O_16_	712.2942	Nominilic acid 17-glucoside
**22**	21.6	477.0685 [M − H]^−^	C_21_H_18_O_13_	478.0747	Quercetin 3-*O*-glucoronide
**23**	22.4	5151195 [M − H]^−^	C_25_H_24_O_12_	516.1268	3,4-*O*-Dicaffeoylquinic acid
**24**	22.9	515.1209 [M − H]^−^	C_25_H_24_O_12_	516.1268	3,5-*O*-Dicaffeoylquinic acid
**25**	23.3	601.1226 [M − H]^−^	C_28_H_26_O_15_	602.1272	3,5-*O*-Dicaffeoyl-4-malonylquinic acid
**26**	23.6	515.1203 [M − H]^−^	C_25_H_24_O_12_	516.1268	4,5-*O*-Dicaffeoylquinic acid
**27**	24.0	601.1209 [M − H]^−^	C_28_H_26_O_15_	602.1272	3,5-*O*-Dicaffeoyl-4-malonylquinic acid isomer
**28**	24.4	601.1228 [M − H]^−^	C_28_H_26_O_15_	602.1272	Eriodictyol 7-(6-galloylglucoside)
**29**	24.7	1019.5149 [M + Formate]^−^	C_49_H_79_O_22_	1019.5063	Asiaticoside B
**30**	24.8	1019.5149 [M + Formate]^−^	C_49_H_79_O_22_	1019.5063	Madecassoside
**31**	25.1	529.1351 [M − H]^−^	C_26_H_26_O_12_	530.1424	3-Caffeoyl-4-feruloylquinic acid
**32**	25.2	873.4522 [M + Formate − H]^−^	C_43_H_69_O_18_	874.4562	Centellasaponin B
**33**	25.7	1003.5166 [M + Formate − H]^−^	C_49_H_79_O_21_	1004.5192	Centellasaponin A
**34**	26.1	957.5088 [M − H]^−^	C_48_H_78_O_19_	958.5137	Asiaticoside
**36**	27.8	1061.5180 [M − H]^−^	C_51_H_82_O_23_	1062.5247	Avenacoside A
**39**	29.8	987.5213 [M − H]^−^	C_49_H_79_O_20_	988.5243	Soyasaponin I
**40**	30.2	571.0881 [M − H_2_O − H]^−^	C_30_H_20_O_12_	572.0955	Manniflavanone

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
