# Peer review of "Structural Stabilization of Human Transthyretin by Centella asiatica (L.) Urban Extract: Implications for TTR Amyloidosis"

_biomolecules, 2019, doi:10.3390/biom9040128_

Round 1

Reviewer 1 Report

Fredrick Nwude Eze et al., identified that hydrophilic fraction of C. asiatica (CAB) has a potential for therapeutics of transthyretin (TTR) amyloidosis. A number of required methods have been applied to prove CAB extract is beneficial; but scientific validation in terms of dosage used in the study was not fully explained and performed. Moreover, chemical identification by QTOF-MS only provides the information of contents, however, responsible molecules were not fully identified and fully discussed. Finally, although authors argue that CAB contains a lot of chemicals as mixture and its strong anti-oxidant activity, safety profiles has not been assessed in this study, which should be important to finally determine if further chemical isolation is required. Overall, results itself provided by authors are convincing, but there are several missing points that can be further explored.

Major point:

1. Despite a many approaches, information on dosage condition and its validity are not well described. Authors could also show more dose-dependent analysis. Authors should kindly indicated in figure legends.

2. Do authors also tried other fractionations? Authors could show the scheme of Acetone/methanol/water extraction procedure and other fractionation if applicable.

3. Chemical identification by QTOF-MS provides important information. There seems to be several dominant components in this fraction, but authors had no further analysis using pure compounds derived from either the extract or commercially available chemicals. These screening can at least tell which is responsible molecules in the extract.

4. Although potential usage of CAB in therapeutic of TTR amyloidosis is convincing. Moreover, authors show the extract has a strong anti-oxidant property in vitro. However, reviewer doubt that CAB extract may have strong toxicity. Safety profile should be important for the future application of the extract.

5. Authors only consider age-based acquired wild-type TTR amyloidosis. Reviewer wonder if the extract is effective to hereditary ATTR amyloidosis such as Met30 mutation.

Minor point:

1. Figure 1 should be combined with Figure 2, or Figure 1 could be shown in supplemental figure.

2. In Figure 2b, statistic analysis should be shown.

3. In Figure 3c, presentation of CAB concentration is unclear. What the number 1 to 5 mean?

4. Figure 4a and 4b, presentation of sample order seems to be parallel. It is not easy to understand.

5. Based on TEM study, CAB affect TTR fibril formation, but does not affect TTR aggregate formation. What is the interpretation from the result?

Author Response

Dear Reviewer,

We would like to immensely thank you for providing critical perspectives and valuable suggestions. Below is our point-by-point responses to the issues raised.

Comments and Suggestions for Authors

Fredrick Nwude Eze et al., identified that hydrophilic fraction of C. asiatica (CAB) has a potential for therapeutics of transthyretin (TTR) amyloidosis. A number of required methods have been applied to prove CAB extract is beneficial; but scientific validation in terms of dosage used in the study was not fully explained and performed. Moreover, chemical identification by QTOF-MS only provides the information of contents, however, responsible molecules were not fully identified and fully discussed. Finally, although authors argue that CAB contains a lot of chemicals as mixture and its strong anti-oxidant activity, safety profiles has not been assessed in this study, which should be important to finally determine if further chemical isolation is required. Overall, results itself provided by authors are convincing, but there are several missing points that can be further explored.

Major point:

1.     Despite a many approaches, information on dosage condition and its validity are not well described. Authors could also show more dose-dependent analysis. Authors should kindly indicate in figure legends.

Response: We are very grateful to the reviewer for these comments. To investigate the effect of a pure compound, the experiment is usually performed at protein to compound ratio of 1:10 (Ferreira et al., 2009). Since partial fractionated plant extract, but not a pure compound, was used in our studied; in addition, the effect of CAB on the quaternary structure stability of TTR was observed (Figure 3c, which is now Figure 2c of the revised manuscript) when the concentration of CAB was 50 to 200 folds greater than that of TTR (in µg/µL). Therefore, TTR to CAB concentration ratio of 1:100 was selected for our experiment. To make it clearer for readers, the explanation of the condition selection was added (line 168 to 170), and the concentration of CAB was informed in the legend of Figure 2 of the revised manuscript. 

2.     Do authors also tried other fractionations? Authors could show the scheme of Acetone/methanol/water extraction procedure and other fractionation if applicable.

Response: We are grateful to the reviewer for this suggestion. Unfortunately, we have not tried other fractionations yet; however, it is in our plan. We agree that the scheme of the extraction procedure will help getting more understanding to our work, so the scheme of the preparation of CAB was included in the Supplementary material as Figure S2 of the revised manuscript.

3.     Chemical identification by QTOF-MS provides important information. There seems to be several dominant components in this fraction, but authors had no further analysis using pure compounds derived from either the extract or commercially available chemicals. These screening can at least tell which responsible molecules in the extract is.

Response: We are very grateful to the reviewer for this comment. We acknowledge that use of pure phytochemicals could provide relevant information on some compounds contributing to the overall effects of CAB on huTTR. There are a lot of pure compounds which have been shown to impair transthyretin amyloidogenesis such as the natural phenolics compounds quercetin, resveratrol and EGCG. These compounds often target a specific aspect of the disease or component of the TTR amyloid cascade. Taking into consideration the fact that TTR amyloidosis is a complex pathology, we are of the view that use of a phytocomplex serve as a multiagent with multitarget potentials. In addition, there is a well-established that the components in a phytocomplex often exert their effect in synergy or addition. While screening for the individual compounds in CAB contributing to the observed effects is the subject of our coming investigation, the focus of this our manuscript was on the anti-amyloidogenic potential “whole” phytocomplex not the individual compounds present as we mentioned (Lines 661-667) in the revised manuscript.  

4.     Although potential usage of CAB in therapeutic of TTR amyloidosis is convincing. Moreover, authors show the extract has a strong anti-oxidant property in vitro. However, reviewer doubt that CAB extract may have strong toxicity. Safety profile should be important for the future application of the extract.

Response: We are thankful to the reviewer for this valuable suggestion and we certainly agree that the safety profile of CAB is very important for its future application. This point has been noted and would constitute a vital aspect in our future investigations.

5.     Authors only consider age-based acquired wild-type TTR amyloidosis. Reviewer wonder if the extract is effective to hereditary ATTR amyloidosis such as Met30 mutation.

Response: We are very grateful to the reviewer for this question. The effectiveness of the extract to other types of ATTR amyloidosis is under investigation in our laboratory. However, because the crucial step in the molecular pathogenesis of ATTR amyloidosis, i.e. dissociation of transthyretin tetramer, is the same in both acquired and hereditary forms of the disease; in addition, the process could be mitigated by CAB. Therefore, similar effectiveness of CAB to hereditary ATT amyloidosis is expected.

Minor point:

1.     Figure 1 should be combined with Figure 2, or Figure 1 could be shown in supplemental figure.

Response: We are thankful to this suggestion. In the revised manuscript, Figure 1 has been moved to Supplementary Materials section as Figure S1.

2.     In Figure 2b, statistic analysis should be shown.

Response: We are thankful to this suggestion. The detail of the statistic analysis was added to legend of Figure 2b (now is Figure 1b) of the revised manuscript.

3.     In Figure 3c, presentation of CAB concentration is unclear. What the number 1 to 5 mean?

Response: We are very grateful to the reviewer for the comments. The additional details of CAB concentration and the sample numbering were added to the legend of Figure 3c (now is Figure 2c) of the revised manuscript.

4.     Figure 4a and 4b, presentation of sample order seems to be parallel. It is not easy to understand.

Response: We are very grateful to the reviewer for the comment. To make it clearer, the additional information was added to the legend of Figure 4 (now is Figure 3) and the labeling of sample was improved.

5.     Based on TEM study, CAB affect TTR fibril formation, but does not affect TTR aggregate formation. What is the interpretation from the result?

Response: We are grateful to this valuable comment.

The results from TEM study could be interpreted that CAB presumably modifies huTTR oligomers such that they are no longer able to proceed to mature fibrils, which is possibly similar to that the observation in some natural products such as EGCG that have been shown to induce the formation of innocuous “off amyloid-pathway TTR oligomers.

Yours Sincerely,

Assoc. Prof. Dr. Porntip Prapunpoj

Department of Biochemistry

Prince of Songkla University, Hat Yai

Songkhla 90112

Thailand

Reviewer 2 Report

Attached

Author Response

Dear Reviewer

We would like to immensely thank you for providing critical perspectives and valuable suggestions. Below are our line-by-line responses to the issues raised.

The reviewer’s comments

1. Eze et al show extracts from Centella asiatica (CAB) stabilize the transthyretin tetramer, preventing its dissociation into the aggregation prone monomer and prevent amyloid formation. Indirect evidence is provided that CAB binds, at least partly, to the T4 site of transthyretin. CAB is also shown to be an anti-oxidant and may mitigate the impact of oxidatively damaged transthyretin. The experiments are well done and I have no issues with the manuscript being published in Biomolecules.

Consider the following optional experiments. Although they are not necessary for publication in my opinion and are purely optional, I believe they have the potential to clarify the mechanism of CAB inhibition:

• Figure 5: Some polyphenols polymerize after autooxidation.1, 2 The resulting structures look fairly similar to the large amorphous aggregates shown in Fig 5.3,4 Is it possible to get an electron micrograph of CAB incubated under similar conditions but in the absence of protein?

Response: We very appreciate the reviewer for this worthwhile suggestion. We currently carried on the experiment as the suggestion (incubating CAB under similar conditions i.e. pH 4.0, 37oC for 7 days). This would take approximately 10 days for the electron micrograph result comes out, figure preparation, and be included in the Supplementary Materials.

2. Localizing the binding site. Based on the NBT results, it is possible that the phenolic components in CAB form a covalent adduct with transthyretin.2,5, 6 One possibility is the formation of a Schiff base5,6 between K15 in the T4 binding site7 and CAS. Acetylating the lysines5,6 would block this interaction and provide a possible mechanism of inhibition. Does acetylated transthyretin bind CAB or prevent the dissociation of transthyretin tetramers?

Response: We are grateful to the reviewer for this comment, and agree that the suggested experiment could provide valuable information on the possible molecular interactions facilitating the binding of CAB to huTTR. However, the information we have provided in the present manuscript is enough to support the possible mechanisms via which CAB stabilizes huTTR tetramers (Line 618-629 of the revised manuscript). More insight of the underlying mechanism is our future investigation plan.

Finally, we would like to thank the referee once more for sparing the time to provide us the useful comments.

Yours Sincerely,

Assoc. Prof. Dr. Porntip Prapunpoj

Department of Biochemistry

Prince of Songkla University, Hat Yai

Songkhla 90112

Thailand

Reviewer 3 Report

good work.

Nothing really to suggest.

Maybe some native speaking english corrections, but these also are not obligatory.

Author Response

Dear Reviewer,

We would like to immensely thank for sparing the time to thoroughly read and provide suggestions for improving our manuscript.

Yours Sincerely,

Assoc. Prof. Dr. Porntip Prapunpoj

Department of Biochemistry

Prince of Songkla University, Hat Yai

Songkhla 90112

Thailand

Round 2

Reviewer 1 Report

Authors answered most of the comments; but there is still some points that needs to be addressed.

Based on the answer, authors felt that extract, or mixture of Centella asiatica (L.) Urban rather than pure single compounds, may be crucial for the effects on TTR stabilization. I understand the points, but if it is the case, safety profile is necessary to confirm in the present study.

The title should be change to "Structural stabilization of human transthyretin by Centella asiatica (L.) Urban extract: Implications for TTR amyloidosis". This is more accurate.

Just curious. Are there any available pure compounds derived from Centella asiatica (L.) Urban, which you identified in the MS analysis?

Author Response

Dear Reviewer,

We would like to immensely thank you for providing further valuable comments and relevant suggestions. Below is our point-by-point responses to the issues raised.

Comments and Suggestions for Authors

Authors answered most of the comments; but there is still some points that needs to be addressed.

Based on the answer, authors felt that extract, or mixture of Centella asiatica (L.) Urban rather than pure single compounds, may be crucial for the effects on TTR stabilization. I understand the points, but if it is the case, safety profile is necessary to confirm in the present study.

Response: We are very grateful to the reviewer for this suggestion. We understand what the reviewer suggested and agree with the reviewer on the necessity of determining the safety profile of CAB particular for any potential future applications. For the present study, the information we provided is enough to support our conclusion. We have planned to adopt a CAB preparation protocol that reflects this concern as well as determining its safety profile in details for a scheduled future investigation of the neuroprotective effect of CAB.  

The title should be change to "Structural stabilization of human transthyretin by Centella asiatica (L.) Urban extract: Implications for TTR amyloidosis". This is more accurate.

Response: We are grateful to the reviewer and agree with this suggestion.

Just curious. Are there any available pure compounds derived from Centella asiatica (L.) Urban, which you identified in the MS analysis?

Response: Yes. Some of the compounds we identified in CAB by LC-MS are commercially available in pure form such as the chlorogenic acids (e.g. 3-O-Caffeoylquinic acid and 5-O-Caffeoylquinic acid) and the terpenoids (asiaticoside and madecassoside).

Once again, we are extremely grateful to the reviewer for finding time to provide valuable suggestions and thoughtful comments on how to improve our manuscript.

Best regards,

Assoc. Prof. Dr. Porntip Prapunpoj

Department of Biochemistry

Prince of Songkla University, Hat Yai

Songkhla 90112, Thailand